# Signal-Based Methods in Dielectrophoresis for Cell and Particle Separation

**DOI:** 10.3390/bios12070510

**Published:** 2022-07-11

**Authors:** Malihe Farasat, Ehsan Aalaei, Saeed Kheirati Ronizi, Atin Bakhshi, Shaghayegh Mirhosseini, Jun Zhang, Nam-Trung Nguyen, Navid Kashaninejad

**Affiliations:** 1School of Electrical and Computer Engineering, College of Engineering, Tehran University, Tehran 14399-57131, Iran; m.farasat66@ut.ac.ir (M.F.); atin.bakhshi@ut.ac.ir (A.B.); sh.mirhosseini@ut.ac.ir (S.M.); 2School of Mechanical Engineering, Shiraz University, Shiraz 71936-16548, Iran; ehsanalaei@shirazu.ac.ir (E.A.); saeedkheirati@shirazu.ac.ir (S.K.R.); 3Queensland Micro-Nanotechnology Centre, Griffith University, Nathan, QLD 4111, Australia; jun.zhang@griffith.edu.au (J.Z.); nam-trung.nguyen@griffith.edu.au (N.-T.N.)

**Keywords:** dielectrophoresis, microfluidics, cell separation, particle sorting, Clausius–Mossotti factor, crossover frequency

## Abstract

Separation and detection of cells and particles in a suspension are essential for various applications, including biomedical investigations and clinical diagnostics. Microfluidics realizes the miniaturization of analytical devices by controlling the motion of a small volume of fluids in microchannels and microchambers. Accordingly, microfluidic devices have been widely used in particle/cell manipulation processes. Different microfluidic methods for particle separation include dielectrophoretic, magnetic, optical, acoustic, hydrodynamic, and chemical techniques. Dielectrophoresis (DEP) is a method for manipulating polarizable particles’ trajectories in non-uniform electric fields using unique dielectric characteristics. It provides several advantages for dealing with neutral bioparticles owing to its sensitivity, selectivity, and noninvasive nature. This review provides a detailed study on the signal-based DEP methods that use the applied signal parameters, including frequency, amplitude, phase, and shape for cell/particle separation and manipulation. Rather than employing complex channels or time-consuming fabrication procedures, these methods realize sorting and detecting the cells/particles by modifying the signal parameters while using a relatively simple device. In addition, these methods can significantly impact clinical diagnostics by making low-cost and rapid separation possible. We conclude the review by discussing the technical and biological challenges of DEP techniques and providing future perspectives in this field.

## 1. Introduction

Particle and cell separation, isolation, and detection are necessary procedures for a wide range of biomedical investigations and clinical diagnostics. Centrifugation, chromatography, gel electrophoresis, flow cytometry, and other conventional separation techniques have been developed to perform these tasks. Large volumes of samples are necessary for these techniques, and sample loss is inevitable [1]. Microfluidics enables the miniaturization of these analytical devices by controlling the fluid flow along a microchannel. Moreover, it is widely used in particle manipulation processes, such as focusing, sorting, and trapping micro to nanoparticles [2,3]. These microfluidic platforms provide fast response time, portability, precise manipulation, low cost, and less sample volume consumption [4].

Particle separation using microfluidic-based techniques is classified into active and passive types. Passive sorting techniques utilize the interaction between the microchannel, particles, and the flow field. Various passive separation methods have been identified, such as inertial and Dean flow fractionation, cross-flow filtration, micro-hydrocyclone, Zweifach–Fung effect, and sorting based on deformability.

In contrast, an external field is used for sorting particles in an active sorting technique [2,3,4]. Dielectrophoresis (DEP), magnetic [3,4,5,6], optical [7,8,9], and acoustic [10] methods are some examples of active sorting techniques. Generally, active sorting methods result in higher efficiency and throughput than passive sorting ones. Thus, passive separation techniques are preferred in critical energy input applications, whereas active separation techniques are selected when higher particle sorting efficiency is required [5].

DEP is the movement of polarizable particles exposed to a non-uniform electric field. It can address multiple particle properties simultaneously, and the behavior of DEP force depends on the particle size, shape, and material. This force can separate particles selectively with no need for particle charge or labeling [11,12]. In 1978, for the first time, Pohl and his student reported the isolation of live and dead yeast cells in inhomogeneous electric fields, which motivated tremendous DEP research efforts [13].

DEP is an interdisciplinary topic from physics to applications. In recent years, DEP and its applications in different fields, such as DEP-on-a-chip systems, have been reviewed in various studies. For instance, the theory and principles of DEP-based manipulation of particles were studied by Cetin et al. [14]. In another review, Pethig [15] described the current status of DEP’s theory, technology, and applications. Maidin et al. analyzed the biomedical applications of DEP, especially in stem cell therapies, liquid biopsies, and infectious diseases [16]. Rahman et al. also investigated the applications of DEP in biomedical science, including eukaryotic and prokaryotic cells, stem cells, oncology, and drug delivery [17]. Pesch and Du focused on DEP applications outside the biomedical scope [11]. In a similar work, Chen and Yuan reviewed the manipulation of polystyrene particles by DEP [18]. Chan et al. [19] and Gascoyne and Shim [20] studied the use of DEP in cancer cells’ manipulation and isolation. Here, we will focus on the developed signal-based techniques in DEP for the cell/particle manipulation that rely on the AC signal parameters.

The integration of microfluidics with DEP has provided the possibility of label-free, fast, inexpensive, selective, and sensitive characterization of target bio-particles. Dielectrophoretic approaches can be coupled with current biological tools for innovative goals such as cell sorting, long-term studies of single cells, fluorescence assessment of cells in small populations, cellular kinetics, and other similar on-chip analyses [21,22]. Lab-on-a-chip (LOC) technology combines small fluid and sample handling techniques with process or detection capabilities. LOC-based systems have been utilized in clinical diagnostics, point-of-care systems, and molecular biology [23,24]. Dielectrophoretic systems can be incorporated into these LOC-based systems to isolate cell groups based on their dielectric properties and trap cells for additional investigations. A variety of DEP-on–a-chip devices have been developed for bioparticle separation so far [21,25,26,27].

This review provides a detailed study of the signal-based methods in DEP for cell/particle separation and manipulation, which rely on AC signal parameters, including frequency, amplitude, phase, and shape. The dielectrophoretic force exerted on the particles/cells is associated with the various dielectric parameters of the particles/cells and their surrounding medium. Hence, cells and particles can be distinguished by changing these parameters based on their dielectric and biophysical features. Signal-based methods allow us to sort and identify the particles/cells by simply modifying signal parameters. Instead of employing time-consuming fabrication methods or complicated geometries, these methods focus on the operating electrical parameters and using a simple device, which usually includes simple planar electrodes and a microchannel. Despite their simplicity, these devices are flexible in configuration, fabrication, and operation. First, the DEP principle and the biological cells’ characteristics are studied. Then, system considerations are provided to help have an optimal situation for separation, like a suitable electrode design. Finally, different methods using these parameters for cell/particle separation are identified.

## 2. Theoretical Background

The interaction of the neutral particle’s dipole moment with a non-uniform electric field’s spatial gradient leads to particle motion. This phenomenon is called dielectrophoresis (DEP). DEP possesses various benefits for working with neutral bio-particles due to its high selectivity, sensitivity, and noninvasive nature. It realizes trapping, translating, focusing, fractioning, and characterizing biological, inorganic, and chemical analytes suspended in a fluid medium. The difference between the polarizabilities of the particles and the suspending medium is the main principle of the DEP, and it can be utilized on any charged or uncharged particles. Imagine a tiny particle in a solution in the presence of an electric field. Charges accumulate at the interface with the medium due to this non-uniform field. This charge accumulation creates dipoles that interact with the field. When the applying field is inhomogeneous on both sides of the particle, the particle experiences a net force. The net force directs to the areas of the high electric field if its polarizability is higher than the medium. However, in the opposite case, it happens in the other direction.

The dielectrophoretic force on a spherical particle is expressed as:(1)$${\vec{F}}_{DEP}=2\pi r^{3}\varepsilon_{0}\varepsilon_{r}Re\left(f_{CM}\right)\nabla E^{2}$$
where $\varepsilon_{0}=8.854\times 10^{-12}F/m$, is the permittivity of the vacuum, and $\varepsilon_{r}$ is the relative permittivity of the medium. *r* is the particle’s radius, *f_CM_* is the Clausius–Mossotti (CM) factor, and *E* is the electric field.

Examination of the above relation shows that the applied force direction relies on the real part of the CM factor. If $Re\left(f_{CM}\right)>0$, the particle is attracted into the areas with the maximum field (Figure 1A(a)); this is called positive DEP (pDEP). On the other hand, (Figure 1A(b)), the particle is repulsed from these areas. In fact, the solvent is pulled toward these areas and causes particle repulsion. This is called a negative DEP (nDEP), where particles seem to be driven into the low-electric field regions [28].

### 2.1. Clausius–Mossotti (CM) Factor

Clausius–Mossotti factor determines the polarity and magnitude of the induced dipole moment in the particle in a non-uniform electric field. This relation is named after the Italian scientist Ottaviano-Fabrizio Mossotti, who investigated the relationship between the dielectric constants of two different media, and the German scientist Rudolf Clausius, who gave the formula in his book [29].

The dipolar CM factor ($f_{CM}$) is given by [30]:(2)$$f_{CM}\left(\varepsilon_{p}^{*},\varepsilon_{m}^{*},\omega \right)=\frac{\varepsilon_{p}^{*}\left(\omega \right)-\varepsilon_{m}^{*}\left(\omega \right)}{\varepsilon_{p}^{*}\left(\omega \right)+2\varepsilon_{m}^{*}\left(\omega \right)}$$

$\varepsilon_{p}^{*}$ and $\varepsilon_{m}^{*}$ are the frequency-dependent particle complex permittivities and its suspension, respectively. Each material’s complex permittivity is $\varepsilon^{*}=\varepsilon -i\frac{\sigma }{\omega }$ (where $i=\sqrt{-1}$) that depends on the electrical conductivity and dielectric constant. Complex permittivity refers to the polarizability of a substance (particle or medium) under an electric field. Both the real and imaginary parts of *f_CM_* are between −0.5 and 1. The imaginary part is associated with the loss (like heat) happening during the polarization [31]. Electronic permanent dipolar, electrical conductivity, and interfacial mechanisms of polarization can contribute to the complex dielectric properties [11,32]. The real and imaginary parts of the Clausius–Mossotti factor are expressed as [33]:(3)$$Re\left(f_{CM}\left(\omega \right)\right)=\frac{\omega^{2}\left(\varepsilon_{p}-\varepsilon_{m}\right)\left(\varepsilon_{p}+2\varepsilon_{m}\right)+\left(\sigma_{p}-\sigma_{m}\right)\left(\sigma_{p}+2\sigma_{m}\right)}{\omega^{2}{\left(\varepsilon_{p}+2\varepsilon_{m}\right)}^{2}+{\left(\sigma_{p}+2\sigma_{m}\right)}^{2}}$$
(4)$$Im\left(f_{CM}\left(\omega \right)\right)=\frac{\omega \left(\sigma_{p}-\sigma_{m}\right)\left(\varepsilon_{p}+2\varepsilon_{m}\right)-\left(\varepsilon_{p}-\varepsilon_{m}\right)\left(\sigma_{p}+2\sigma_{m}\right)}{\omega^{2}{\left(\varepsilon_{p}+2\varepsilon_{m}\right)}^{2}+{\left(\sigma_{p}+2\sigma_{m}\right)}^{2}}$$

The first limit for the real part is regarded at low frequencies and creates an ionic contribution to the conductivity [33]:(5)$$Re\left(f_{CM}\left(\omega \to 0\right)\right)=\frac{\left(\sigma_{p}-\sigma_{m}\right)}{\left(\sigma_{p}+2\sigma_{m}\right)}$$

It is simple to verify that this limit can be positive if $\sigma_{p}>\sigma_{m}$ or negative if $\sigma_{p}<\sigma_{m}$. Another limit is considered at higher frequencies, giving the ionic contribution to the permittivity [33]:(6)$$Re\left(f_{CM}\left(\omega \to \infty \right)\right)=\frac{\left(\varepsilon_{p}-\varepsilon_{m}\right)}{\left(\varepsilon_{p}+2\varepsilon_{m}\right)}$$

The frequency response of these two cases for some given input parameters is shown in Figure 1B [14]. It is easy to verify that this limit can be positive if $\varepsilon_{p}>\varepsilon_{m}$ or negative where $\varepsilon_{p}<\varepsilon_{m}$. As mentioned in the previous section, in positive amounts of *f_CM_*, the particle experiences a DEP force that leads it to the high electric field areas because the induced dipole moment is directed in the same direction as the electric field. In contrast, in the negative values, the particle feels a DEP force that leads it away from the high field areas because the induced dipole moment is directed in the opposite direction to the field. Where $\varepsilon_{p}=\varepsilon_{m}$, this factor is zero. Hence, the particle is not polarized. The critical applications of this zero-polarization situation include determining the dielectric properties of the particle, and monitoring the precise changes of particle dielectric properties (for example resulting from cell differentiation or apoptosis). Moreover, this situation can be used for particle separation in a mixture in which the condition $\varepsilon_{p}=\varepsilon_{m}$ for each different particle happens at a different frequency [29]. It is possible to detect the crossover frequency ($\omega_{c}$) for which *f_CM_* is zero and DEP goes from a positive to negative situation or vice versa [33]:(7)$$\begin{array}{c} Re\left(f_{CM}\left(\omega_{c}\right)\right)=0\\ \;\omega_{c}=\sqrt{\frac{\left(\sigma_{m}-\sigma_{p}\right)\left(\sigma_{p}+2\sigma_{m}\right)}{\left(\varepsilon_{m}-\varepsilon_{p}\right)\left(\varepsilon_{p}+2\varepsilon_{m}\right)}} \end{array}$$

### 2.2. Cell Characteristics

Different cell groups usually have different characteristics. Differences in size, conductivity, membrane capacitance, and other properties have resulted in different dielectrophoretic behaviors of particles and cells. Changes in the cells’ dielectric properties are associated with their physiological, biochemical, and morphological changes. Cell types with various surface areas and size characteristics show different DEP frequency responses, and by choosing a suitable field frequency between their crossover frequencies, we can separate them. In this case, cells with lower crossover frequencies are attracted to high field regions, while those with higher crossover frequencies are repelled from low field regions. Gascoyne and Shim [20] characterized the NCI-60 panel of cancer cells and exhibited that all solid tumor cell lines have distinct crossover frequencies from normal blood cells.

Two influential dielectric parameters in the cells are membrane capacitance and interior conductivity. Since membrane capacitance affects the experienced force at a given frequency, this value determines the dielectrophoresis differences in various cell types. Therefore, cells with different membrane capacitances can be separated using dielectrophoresis [34]. Changes in the membrane capacitance are often closely related to cell differentiation processes [35]. For instance, it can distinguish human embryonic stem cell (hESC) derivatives after differentiating. The capacitance of the membrane increases as the cells differentiate into a phenotype similar to mesenchymal stem cells. In addition, Menachery et al. showed that small cell lung cancer (SCLC), which has an adherent phenotype, has an increased membrane capacity relative to the phenotype with mainly suspension characteristics. These changes may be related to the modification of the cell surface by bound neural cell adhesion molecule (NCAM) and polysialic acid (PSA) [34]. Moreover, the difference in membrane capacitance can reveal the cell surface morphology. For example, the cell membrane morphology of oral squamous cell carcinoma (OSCC) has been studied by DEP and scanning electron microscopy (SEM). It was found that cells with higher tumor-forming potential had higher membrane effective capacitance, a rougher surface, and abundant cell protrusions [36].

The dielectric characteristics of cancer cells have been investigated in various phenotypes of cancer and used in distinguishing them. Many reports show that the membrane’s effective capacitance in cancer cells increases relative to the cells with a more usual phenotype. Changing the membrane morphology (cancer cells seem to have a rougher surface with folds, ruffles, and microvilli) may be the reason. In most cell groups, especially in cancer cells, the plasma membrane is not flat and contains small and large features, including folds, microvilli, and ruffles. Hence, mammalian cells have larger membrane surface areas than idealized, flat spheres with the same volume [30,37].

Gascoyne and Shim [20] showed that cancer cells have a 50% to 300% larger capacitance per unit area than normal phenotype cells. In addition to the folding factor of the membrane, cancer cells have a larger radius (R) than their normal counterparts [34]. Both factors are influential in dielectric phenotype differences among normal and cancer cells. Cell adopts a volume and membrane area that conforms to its surrounding cells in a tissue. However, cytoskeletal tension leads to a pseudo-spherical form for the cell when released into a suspension. Moreover, as tumor grade increases and cancer progresses, cell membrane areas become more significant than normal cells [38,39]. Suppose cells that generally grow in contact with other cells are maintained in suspension for a long time. In that case, they usually undergo cytoplasmic changes by which excess cytoplasm and membrane are shed by large vesicles [40]. This process causes a cell size reduction while the cell membrane’s folding factor remains high [40], and the cancer cells maintain a distinct dielectric phenotype from the blood cells. It suggests that DEP can isolate circulating tumor cells from the blood cells even when they show similar cytoplasmic and membrane shedding in circulation and have similar sizes to blood cells after leaving their origin tumors [20].

Interior conductivity value is related to different physiological phenomena, such as apoptosis. For instance, live and dead cells can be separated by the difference in their conductivity. In fact, after the cell death, the cell membrane permeability increases, resulting in its higher conductivity. Moreover, McGrath et al. showed that PDAC (Pancreatic ductal adenocarcinoma) cells of higher tumorigenicity exhibited a lowered interior conductivity [41]. In another study, Trainitoid et al. indicated that in mouse ovarian surface epithelial cell line (MOSE) conductance and capacitance of the membrane and conductivity of the cytoplasm increase with increasing the malignant phenotype [42]. Mahmoud Al Ahmad et al. showed that cancer cells with different origins have distinct electrical parameters. Cancer and normal cells presented higher dielectric amounts in the following order (from lowest to highest): breast, lung, and liver [43]. Interior conductivity also can be utilized as an index for testing the sensitivity of drugs in the cells.

In recent years, various techniques have been utilized to characterize the different types of cells, specifically cancer cells. For example, Electrorotation (ROT) and DEP techniques have enhanced single-cell analysis [44]. The magnitude and direction of forces exerted on the cells in these methods are dependent on the frequency and the cells’ intrinsic electrical properties that depend on the cells’ constitution, morphology, phenotype, and structural organization. Thus, characterization methods have also been used to investigate cellular alterations such as mitotic stimulation, gene expression levels, post-translational modifications, environmental effects, glycosylation variations, lipid composition [45], and induced differentiation [34,37,46,47]. The ROT cell characterization technique directly measures the cell parameters [48]. In ROT, shifting the electric field phase produces a rotating field, and the cells’ rotation rates depend on the applied field frequency. As a result, the cells’ dielectric properties are obtained from the frequency dependence of the rotation rates or conductivity dependence of the crossover frequencies [49]. For example, one representative experiment is rotating a single cell at a speed and direction that depends on the field frequency to determine the crossover frequency. The frequency, in which the direction of the cell rotation alters, is the crossover frequency. Becker et al. used this method to determine the crossover frequencies and dielectric parameters of HL60 leukemia cells, T lymphocytes, and red blood cells (RBC) [47]. The different crossover frequencies of these cells show considerable differences in their electrophysiological properties lines; hence, they can be successfully separated from the blood cells [47].

## 3. Design Considerations

### 3.1. Integration of Microelectrodes

The integration of the microelectrodes generates the electric field non-uniformity needed for DEP applications into the microfluidic chip. The field of non-uniformity production geometries can be categorized into two main groups. Electrode-based planar geometries, which are highly effective for field coupling but extend over a limited height of the channel; or insulator-based geometries that extend over the whole channel depth, but show weak field coupling because the electrodes are distant from the field non-uniformities [50,51,52,53]. Most of the signal-based methods introduced here rely on the former group. In this group, a popular approach is to utilize conductive microelectrodes, which are usually in direct contact with the suspension and sample. The common materials for microelectrodes are carbon and Indium tin oxide, as well as metals, including gold, titanium, platinum, and silicon. The stable electrochemical property makes gold and platinum the best choices for these DEP systems. Electrodes are usually fabricated by planar metal using deposition, photolithography, and patterning procedures. Electron beam evaporation or sputtering techniques are used to deposit metal layers. Photolithography followed by wet/drying etching or lift-off techniques is standard for patterning the metal layers. Considering constant properties for particle and medium, the particle movements are mainly associated with the square of the gradient of the applied field (Equation (1)). This term has the unit V^2^/m^3^. The m^3^- dependence of this term illustrates that for a strong force, the distance between the electrodes has to be small [11].

The DEP electrode configurations mostly use simple planar interdigitated electrodes, Figure 2A, on the bottom of the microfluidic channel resulting in a vertical direction of DEP forces. The resulting DEP forces generated by this type of electrodes typically move the particles or cells to different heights of the microchannel. In addition, trapezoidal planar electrodes [54] Figure 2B, and slanted electrode arrays [55] Figure 2C, have been identified for separating different-sized particles in a flow-through channel. Spiral electrodes are used in travelling-wave DEP methods. Figure 2D shows a design of a three-phase electrode array used to produce an electric field phase gradient. These looping electrodes are connected to signals with the same magnitude and frequency but with a phase shift (120° in this case).

For thin-film planar electrodes, typically, there will be dead electrical field space far away from the microchannel bottom [56]. For having a strong electric field covering the whole channel, planar electrodes on both the top ceiling and the bottom of the microfluidic channel can be utilized [57]. In this method, DEP forces can be generated along the width of the channel to deflect the particles transverse to the flow stream and parallel to the substrate. Moreover, using the sidewall electrodes extending along the entire height of the channel, Figure 2E, a gradient of the electric field can be generated uniformly in the direction of the channel height [58]. Another possible method for having a strong field in the whole channel is using 3D electrodes producing an electric field gradient in the height direction inside the channel. Various 3D metal electrode patterning methods have been utilized to boost the electric field extent across the channel height [59] while maintaining the high coupling of the field. However, the fabrication of these geometries has various challenges, including the need for labor-intensive interlayer alignment and highly specialized deposition techniques accessible only in limited facilities [53]. A new approach for 3D metal geometries facile fabrication in the microchannel relies on the co-fabrication of adjoining electrodes and channels [60]. In this method, the liquid metal alloy solidifies in the electrode channel at room temperature [61]. The main advantage of this method is creating patterned metal sidewall electrodes across the entire device height, needless to alignment between the metal electrodes and microchannel layer [53]. Furthermore, this method can prevent contact between the electrodes and cells. For example, Huang et al. [53] have patterned metal over the whole depth (50 μm) of the channel sidewalls using a single lithography step (Figure 2F).

In another technique, 3D conducting PDMS composites have been utilized. These composites are synthesized by mixing silver powders with PDMS gel, and these composite electrodes can readily be integrated with the PDMS microchannel (Figure 2G). Moreover, these sidewall electrodes allow DEP force to distribute three-dimensionally, therefore boosting DEP force effect in the whole channel [62]. In another approach, field non-uniformities are created by media conductivity gradient. This approach enables cell separation based on their iso-dielectric point at the crossover frequency [63].

For different signal-based methods disused in the next sections, various configurations of electrodes have been utilized (Table 1). However, some of the introduced configurations are more usual for these applications. For example, for the crossover-based separation methods studied here, the usual structure is the planar electrodes in different shapes, like two and three-electrode structures. For twDEP, scientists have used Simple interdigitated electrodes, top-bottom electrode arrays, spiral electrodes, and gradient electrodes. In most of the time-varying methods, simple interdigitated electrodes and slanted interdigitated electrodes are common. For moving DEP and FFF-DEP used structures are top-bottom electrode arrays and simple interdigitated electrodes, respectively.

Various novel electrode structures have been identified in different studies to improve the separation process of different types of particles and cells. For example, in Figure 2H, Jino Fathy et al. have proposed new electrode structures to separate viable and non-viable yeast cells by twDEP. These structures are made of chevron-shaped electrodes with a curvy shape at the center, replacing the sharp angle that usually exists at this point. Indeed, they removed this sharp angle to eliminate the irregular electric field and unpredictable DEP and twDEP behaviors of the cells at this place [76]. Jiang et al. (Figure 2I) introduced a new type of electrodes, floating electrodes (FE), to separate yeast cells and PS colloids to eliminate the need for external Ohmic connection to individual array units [77].

Some other methods for producing electric field gradients inside the channel can be used for various DEP applications. For example, optical DEP (oDEP), Figure 2J, is another promising DEP technique developed to manipulate cells. Virtual electrodes are shaped on the photoconductive surface when illumination is emitted on a photoconductive surface; hence, the non-uniform electric field is generated by direct optical images. By this technique, any shape of electrodes can be formed on the photoconductive surface.
Figure 2A variety of electrode configurations used in DEP applications. (**A**) Simple planar interdigitated electrodes; (**B**) planar trapezoidal electrodes; (**C**) planar slanted interdigitated electrode arrays; (**D**) spiral electrodes are used in travelling-wave DEP to produce an electric field phase gradient (Reprinted with permission from [78]. Copyright 2009, Springer); (**E**) side-wall electrodes for having a uniform electric field in the direction of the channel height (Reprinted with permission from [58]. Copyright 2020, Multidisciplinary Digital Publishing Institute (MDPI)); (**F**) 3D electrode geometry created by Co-fabrication of adjoining electrodes and channels (Reprinted with permission from [53]); (**G**) 3D conducting PDMS composites are generated by mixing silver powders with PDMS gel (Reprinted with permission from [62]); (**H**) different chevron electrode structures to eliminate the irregular electric field and unpredictable DEP and twDEP behaviors of the cells. (a) Simple chevron electrodes. (b) Chevron electrodes after removing the sharp angle. (c) Transferring the curve of the electrodes to a side instead of the center. (d) The reverse boomerang-shaped electrodes. (e) Wave-shaped electrodes (Reprinted with permission from [76]. Copyright 2015, Springer); (**I**) Floating electrode arrays eliminate the need for external ohmic connection to individual array units(Reprinted with permission from [77]. Copyright 2018, American Institute of Physics (AIP) Publishing); (**J**) Optical DEP to form any shape of electrodes on -the photoconductive surface (Reprinted with permission from [79]. Copyright 2005, Springer Nature).
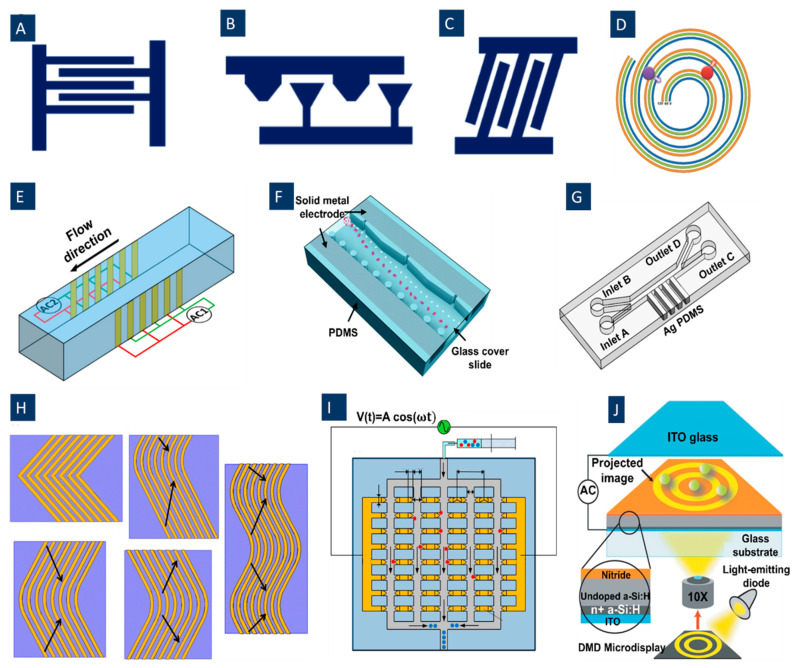


### 3.2. Minimizing the DEP Effects on the Live Cells

Applying electric fields to live cells while working with them can influence them. Theoretically, these influences include at least two aspects: direct interactions of the cell and the field and Joule heating [80].

Applying an electric field into a conductive medium creates power loss in the form of Joule heating in it. Consequently, if the resulting temperature change is more than a few degrees Celsius, it may affect the cell physiology phenotype [80]. Steffen Archer et al. exhibited [81] that after 15 min exposure of an electric field with a voltage of 21V_p-p_ and a medium conductivity of 10 mS/m, just an overall steady-state temperature increase of about 1 °C happened in the medium. This small value is not so harmful to the cells; however, this slight temperature fluctuation can cause up-regulation of heat shock proteins (HSPs). For frequencies above 100 MHz, significant strength is observed in the cells due to the thermal effect. The wasted energy increases with rising frequency, and media and cells absorb this energy [82]. Moreover, it has been found that this induced heating corresponds to the medium conductivity, so a lower conductivity can effectively minimize the temperature excursion and its consequences.

Another possible harm is membrane rupture owing to a high transmembrane voltage (∆∅). The cell membrane may be damaged if transmembrane potential (induced by the electric field) overpasses the membrane dielectric breakdown voltage [80]. In other words, in high-potential electric fields, temporary pores are created on the cell membrane (electroporation), or even the cells can be irreversibly disrupted (electrical lysis). Where electrical membrane lysis occurs, the balance between the osmotic pressure of the cytosol and the outside medium is lost; hence, over-swelling causes the cell lysis [80]. The suitable threshold potential for cancer cells’ membrane is about 1 V [80]. Steffen Archer et al. showed [81] that by increasing the frequency, induced membrane potential decreases. For instance, in a frequency of 5 MHz for a voltage amplitude of 21 V_p-p_ and a medium conductivity of 10 mS/m, the cell membrane potential was about 8.6 μV_p-p_ to 68 μV_p-p_. In contrast, the membrane potential is much higher at lower frequencies, near 5 V_p-p_, enough for cellular membrane rupturing. At very high frequencies (e.g., >100 MHz), the cells’ potentials decline to a negligible amount, making the cell transparent to the electric field [80].

When an electrical field is applied to a cell, some stresses act on its surface. These electrical stresses play an essential role in structural changes of the cell membrane that may cause cell permeabilization. These stresses are associated with the cell and suspension dielectric characteristics and electric field characteristics. Some experimental results illustrate that this phenomenon cause pore formation in the lipid bilayer membrane [80,81,83].

In Figure 3A, the cell surface electric stress time average is depicted at two different frequencies of 10 kHZ and 10 MHz. It shows that the non-uniform electric field influence on the cells is dependent on its frequency. Kia Dastani et al. showed that cell elongation rises with an increasing frequency (500 kHz to 1 MHz). They illustrated that the maximum deformation for the cells (RBCs) occurred at the peak of the Clausius–Mossotti factor versus frequency plot (1 MHz). At higher frequencies (15 and 20 MHz), cell deformation decreased. In Figure 3B, we can also see that in nDEP regime (10 kHz), the electrical stresses have the form of tensile forces, whilst in pDEP regime (10 MHz), compressive stresses are applied over the cell surface [84].

Figure 3C–F show the effects of suspension dielectric properties on the electric stresses’ magnitude at two frequencies of 10 kHz and 10 MHz. These plots show that the electric stresses’ magnitude increases with increasing the ratio of $\varepsilon_{m}/\varepsilon_{p}$ and declines with increasing the ratio of $\;\sigma_{m}/\sigma_{p}$. We can also see that at higher frequencies, the medium conductivity has no influence on the electric stresses [84].

Moreover, living cells’ subjection to a non-specific environment may lead to their genetic modification. Intact cells can perceive these environmental changes as “hits” to their genetic integrity. However, the cells do not experience impressive changes while subjected to an electric field. One of the advantages of using DEP is that it does not apply significant genetic transformations to the cells. For instance, Vahé Nerguizian et al. showed that exposure of MDA-MB-231 cells to 10-kHz (for nDEP) and 100-kHz (for pDEP) fields alter most of their genes. For example, in nDEP, apoptosis is down-regulated, and rRNA transcription is up-regulated. On the other hand, pDEP downregulates energy production and cellular respiration and up-regulates some cellular transcriptional activities [85,86].

Some experimental results show that an influential parameter on the vesicles’ evaporation and, generally, cell transformation in an electric field is the exposure time of the electric field. In other words, longer DEP processes have a higher impact on the cells. Salipante et al. illustrated that vesicles could be ruptured in a weak electric field applied for an extended period. At the same time, they survive in a strong electric field used for a short duration [87]. Most of the current studies carry out DEP for short times. Prior studies show that an exposure of 15 min does not significantly impact the cells and would not significantly influence the cell cycle [81]. However, several living cell types can survive relatively long periods after exposure to relatively strong electric fields for up to 48 h. This has been shown for red blood cells (RBCs) [88], mouse fibroblasts [89], and bacteria and yeast [90].

## 4. Signal-Based Methods

### 4.1. Separation Based on Crossover Frequency

#### 4.1.1. Simple Single-Frequency Methods

Dielectric characteristics of the different particles and cells can be utilized to identify and isolate them. The behavior of different particles varies in an electric field. If particles are more polarizable than the suspending medium, they experience positive DEP (pDEP); otherwise, they feel negative DEP (nDEP). Crossover frequency or Critical frequency is a noteworthy feature in the Clausius–Mossotti (CM) curve of the particles or cells describing the conversion from pDEP to nDEP. This frequency is defined by the dielectric properties of the particles and the surrounding medium, and can be used to identify and manipulate particles and cells [91,92]. IT has been utilized to separate different types of particles and cells, including live and dead cells, cells and PS particles, and different types of cells. To this end, various methods have been used, like shifting the frequency to find the desired separation frequency, changing the medium’s conductivity, or both. For example, in Figure 4B, Zhao et al. used this method to separate the live and dead yeast cells by shifting the ac electric field frequency and finding a desirable separation frequency [92].

The DEP-induced behavior of the biological cells is determined by their extracellular biomarkers, intracellular events, and surface morphology. There are two intrinsic relaxation frequencies in suspension for a single cell. At low frequencies, nDEP force is exerted on the cells. In this situation, the determining factors in the *f_CM_* are just the electrical conductivities of the cell and the medium. At higher frequencies and after the first crossover frequency, the cell membrane capacitance is short-circuited due to the polarization between the cell and the media. Hence, the second relaxation occurs and creates positive DEP. After the second crossover frequency, DEP force is negative again, because the dielectric permittivity of the cell is relatively lower than the suspending medium [92]. In Figure 4A, we can see frequency spectra of the real part of the CM factor for some of the blood cells [93].

This DEP behavior in cells has been employed in different studies to separate them, specifically, the separation of cancer and other types of blood cells. For example, breast CTCs can be successfully isolated from blood cells [94]. The separation of two types of cells is challenging because they may have cross-over frequencies close to each other For a given media conductivity. However, by changing the media conductivity, this difference can be enhanced. For describing the relationship between the first cross-over frequency, the capacitance of the membrane, particle radius, and the conductivity of the medium, the following equation can be utilized [64]:(8)$$f_{x01}=\frac{\sqrt{2}}{2\pi rC_{m}}\sigma_{m}$$
where $C_{M}$ is the capacitance of the cell membrane, r is particle radius, and $\sigma_{m}$ is the conductivity of the medium. Since different types of cells have different values of $C_{M}$, changing the media conductivity can help in increasing their $\;f_{x01}$ difference. In Figure 4C, Alshareef et al. introduced this method to separate MCF-7 and HCT-116 cells [64] with an enrichment efficiency of about 93%.
Figure 4(**A**) Re(*f_CM_*) at different frequencies for blood cells(Reprinted with permission from [93]. Copyright 2017, IJERT.ORG); (**B**) DEP’s separation of live and dead yeast cells is based on finding a good separation frequency(Reprinted with permission from [92]. Copyright 2019, American Chemical Society); (**C**) MCF-7 and HCT-116 cells are separated by changing the medium conductivity to increase their crossover frequencies difference (Reprinted with permission from [64]. Copyright 2013, American Institute of Physics (AIP) Publishing).
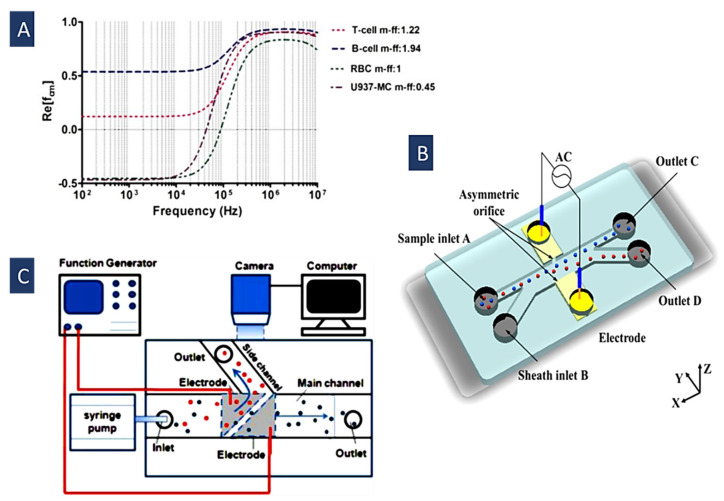



#### 4.1.2. Multiple Frequency Dielectrophoresis (MFDEP)

Using the combination of frequencies in the multiple frequency DEP method enables us to separate particles and cells according to their dielectric properties [66]. This method can be utilized for the selective isolation of particles with very close Clausius–Mossotti spectra, making their separation with only one frequency challenging. In other words, adding every frequency adds up to two additional control parameters for the separation [66]. The performance of DEP devices for manipulating the particles can be improved using multiple frequencies. In an MFDEP system, the inflow and outflow drag forces are substituted by p- and n-DEP forces. In addition, we can apply forces on desired particles according to the difference in their CM factor responses (e.g., by using the frequencies at which CM for some groups of the particles is 0) [66]. Consequently, more groups of the particles can be independently isolated using MFDEP. In Figure 5A, Urdaneta, et al. introduced the idea of isolating different particles using multiple frequencies [66].

As mentioned in the previous section, particles and cells can be conveniently isolated using a frequency that results in an opposite-sign CM factor for each group of particles. To this end, different CM crossover frequencies are necessary for each group of particles. It is desired to have significant differences among the crossover frequencies to compensate for uncertainties (due to variations in the conductivity of the medium, size of the particle and frequency) in the calculated crossover frequencies. Another method is defining an adequate amount for CM factor as follows [66]:(9)$$\vec{\;F}\propto f_{CM}^{eff}\nabla {\left|\vec{E_{t}}\right|}^{2}$$

This effective factor contains the necessary information on the frequency of use in MFDEP. It determines the crucial parameters from multiple frequencies applied to a particle, including the sign of the force and the crossover frequency [66].
(10)$$\bar{f_{CM}^{eff}}=\frac{\left|\sum_{i}^{n}f_{CM,i}\nabla {\left|\bar{E_{i}}\right|}^{2}\right|}{\left|\sum_{i}^{n}\nabla {\left|\bar{E_{i}}\right|}^{2}\right|}$$
where $E_{i}$ is the root mean square (rms) of the applied electric field at frequency $f_{i}$, and $f_{CM}^{eff}$ is a weighted average of the $f_{CM,i}$. Weighting is defined by the gradient of the squares of the electric field. Equation (10) shows how adding each frequency, can add control parameters. $f_{CM}$ is just dependent on the frequency, while, $f_{CM}^{eff}$ depends on both the frequency and the field’s gradient at each frequency [66]. In the cases where just one frequency is utilized, $f_{CM}^{eff}$ can be conveniently reduced to $f_{CM}$. As a result, $f_{CM}$ describes the relative polarization when a single frequency is used, while $f_{CM}^{eff}$ provides the resulting polarization of all the frequencies. Equation (10) can be used for any number of frequencies and any number of electrodes.

Figure 5D depicts how separation takes place in this method. The values for CM were calculated by the single-shell model in a uniform electric field [95]. First, in range a, there is a small window for the isolation of these cells using just one frequency. Then, after adding a 10 kHz signal with a relative amplitude of 0.6 to the first signal, these curves take on different signs on a broader frequency range (range b). In other words, a 10-kHz signal causes a force component used to offset the net force applied to the particle owing to the original frequency; therefore, it enables us to manipulate the crossover frequencies [66].

Unlike $f_{CM}$, $f_{CM}^{eff}$ is a function of location when multiple frequencies are applied by different electrodes. Hence, plots of $f_{CM}^{eff}$ as functions of the position are necessary for forecasting the equilibrium positions of various groups of particles on the geometry of electrodes [66]. Using these plots as functions of location and choosing suitable frequencies and electrode geometry, more types of particles can be isolated. The equilibrium location of the particle can be controlled using the applied signals with different frequencies and, it is determined by the dielectric properties of the particle and the medium.

In Figure 5B, N. Demierre et al. showed how to concentrate a stream of particles into a tunable position across the channel by various frequencies and potentials. More importantly, yeast cells and polystyrene beads can be separated based on their different dielectric responses to multiple-frequency signals [65]. Focusing the particles is conducted by a suitable geometrical arrangement of metal electrodes and insulators, making the opposing DEP forces for focusing the particles and cells possible. After focusing, three signals with different frequencies are combined to separate yeast cells and beads with varying equilibrium positions [65].

In Figure 5C, Ana Valero provided a sorting method based on the internal structure and characteristic shape changes in the yeast cell cycle [96]. Indeed, their method relies on the geometry of the cell rather than its volume. In other words, there is an equilibrium between DEP forces produced by different-frequency electric fields in this method. These fields act on the cells from both sides of a sorting channel in which they are flowing. This force opposition removes the first-order dependence of the force on the cell volume and increases the sensitivity to the variations of the cell shape in the division cycle. As a result, cells with different stages of the division, irrespective of the flow rate, are pushed to the characteristic equilibrium positions [96].
Figure 5(**A**) Separation of live and dead yeast cells using their CM_eff_ in multi-frequency DEP (Reprinted with permission from [66]. Copyright 2007, John Wiley and Sons); (**B**) focusing and separation of PS particles and yeast cells using different sets of potentials and frequencies (Reprinted with permission from [65]. Copyright 2008, Elsevier); (**C**) on-chip cell sorting using their characteristic equilibrium positions for different division stages of yeast cells (Reprinted with permission from [96]. Copyright 2011, Royal Society of Chemistry); (**D**) CM spectra for human T- and B-lymphocytes as a function of frequency with their CM_eff_ after adding a 10-kHz signal with a relative amplitude of 0.6 to the original frequency. The isolation range is widened from a to b (Reprinted with permission from [66]. Copyright 2007, John Wiley and Sons).
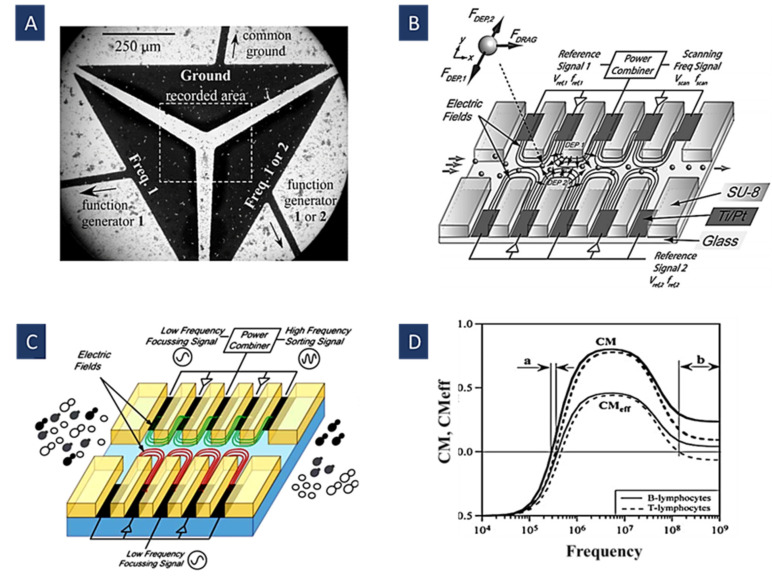



#### 4.1.3. Ultra-High-Frequency Dielectrophoresis (UHF-DEP)

In UHF-DEP, using high frequencies enables us to bypass the cell membrane. Consequently, dielectrophoretic force is more strongly linked to the intracellular properties [59] than the cell membrane properties. This technique enables us to screen a new discrimination parameter for cells, the physical properties, and intracellular differences. This method does not need any labeling and does not affect the viability and integrity of the cell. There are two crossover frequencies *f_x_*_01_ and *f_x_*_02_, in each particle or cell’s CM factor spectral plot for which F_DEP_ becomes zero [97]. Cell parameters affect these frequencies. For instance, the first crossover frequency (*f_x_*_01_) is usually observed in the range of kHz and is affected by the cell membrane properties, size, and shape. The second one (*f_x_*_02_) is typically in the range of MHz in the medium with low conductivity, and is dominated by the internal parameters of the cell, such as cytoplasmic conductivity and permittivity. Choosing the DEP frequency range is essential based on the type of cell properties one wants to access.

If we want some information about the characteristics of the cell plasma membrane, conventional DEP frequencies (usually in the range of 100 kHz to 5 MHz) are useful for cell analysis. The shape, size, and morphology of the cell strongly influence the interaction of the cell with the electric field in this low-frequency regime. Conversely, for providing information about intracellular properties, ultra-high frequencies DEP (from 50 MHz to 500 MHz) will be better [98]. In fact, with increasing the frequency above several tens of MHz, the electric field penetrates the cell through the plasma membrane and interacts directly with the cell interior. Hence, the effect of the generated DEP forces (attractive or repulsive), may be different at high-frequency regimes based on the dielectric properties of the cell content [98].

Crossover frequency can be obtained using Equation (7). Understanding the behavior of cells at the second crossover frequency (*f_x_*_02_) needs a model incorporating their intracellular structure and dielectric properties, mainly by conductivity and to a small amount by permittivity [99]. For instance, undifferentiated cells show various biological characteristics or physiological mechanisms according to their differentiation state; hence, they have different crossover frequencies with differentiated ones. As a result, studying their dielectrophoresis behaviors at this ultra-high frequency regime seems useful for some targeted applications like their separation [98].

For a high throughput cell sorting, cell trapping by pDEP during the isolation has to be prevented because it can cause the formation of the cell agglomerate that can disturb the flow and change the efficiency of the sorting, or even generate clogging inside the microchannel [97]. To this end, a frequency higher than or close to the second crossover frequency (*f_x_*_02_) of the target cell group has to be chosen to apply only nDEP to the cells. For example, Provent et al. [97] utilized this method to separate subpopulations of mesenchymal stem cells (MSC). In fact, two different populations of the cells can be separated by setting the frequency above the highest median crossover frequency value among both cell groups.

A whole population of the cells can have different dielectric features from the other groups due to the natural heterogeneity of the cells and it causes dispersion of the crossover frequencies. Using such dispersion, we can see that all the cells in one group are submitted to the strong nDEP force leading to a strong repulsion. Meanwhile, most of the cells in the other group may just experience very weak DEP forces that cause a limited deviation in their trajectory. Crossover frequency-based methods’ parameters have been reported in Table 2.

### 4.2. Travelling Wave Dielectrophoresis (twDEP)

Traveling-wave electric fields can be created by applying phase-shifted voltages to parallel electrodes. The particle under the field of travelling waves will move along or against its direction of travel [100]. In the early 1990s, twDEP was developed to improve sorting and trapping accuracy, controlling motion and selectivity. Longer, multiple discrete electrodes are sequentially powered with signals of shifted electric phases in the twDEP setup. The DEP force’s magnitude leads to particle parallel displacement along the microelectrodes [101].

As shown in Figure 6A, the key point is that travelling-wave electric fields provide continuous displacement of particles. Despite other methods for cell separation, which generally need a continuous flow, a complicated and big footprint network, and an external pump; twDEP method would be able to work with a wide range of particles without using any external pumping sources to produce particle motions horizontally and separate them [102].

Figure 6B shows that sorting could be achieved with a sustained twDEP particle force normal to the continuous flow applied over the whole device by a single 3D electrode array. Microparticles can be continuously fractionated into various downstream sub-channels depending on their twDEP mobility on either side of the cross-over [67] by applying this system. Figure 6C illustrates a twDEP-based cell sorting device that utilizes two different electrode arrays. This design applies focusing force to a continuous flow to increase the velocity approximately fourfold.

A twDEP system was found to perform better than DEP trapping in terms of throughput. However, it requires complicated electrode structures, making it challenging to improve the recovery rate and reproducibility [68].
Figure 6(**A**) Particle exposed to twDEP; (**B**,**C**) schematic diagrams of continuous high-throughput particle sorters. These methods are based on 3D twDEP separation of microparticles into different downstream sub-channels according to differences in their twDEP mobilities (Reprinted with permission from [67,68]. Copyright 2009, Royal Society of Chemistry and Copyright 2010, Springer, respectively); (**D**) separation of cells and bacteria by twDEP using gradient electrode (Reprinted with permission from [69]. Copyright 2011, Elsevier); (**E**) separation, trapping, and channeling of microparticles based on their physical parameters using twDEP (Reprinted with permission from [103]. Copyright 2009, Institute of Physics (IOP) Publishing).
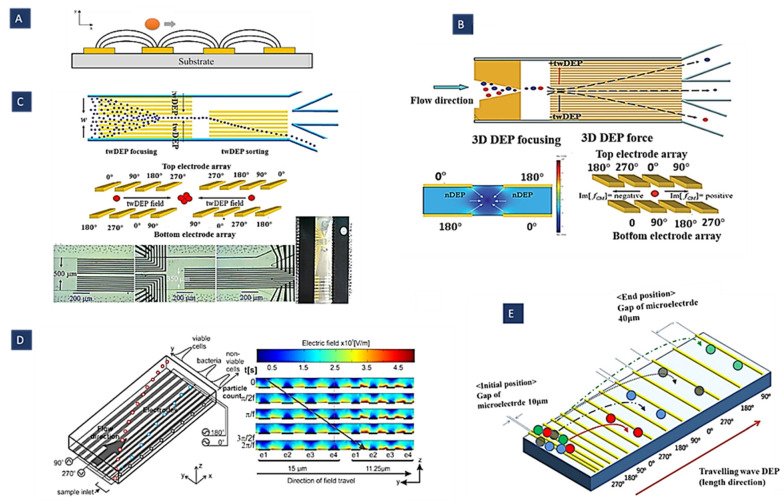



In Figure 6D, an array of parallel electrodes with increasing gap and width size was used along the microchannel to expose cells to a pressure-driven twDEP force. A mixture of L. casei bacteria and viable S. cerevisiae cells is separated into distinct populations by creating a barrier bacteria cannot cross using this design. Indeed, as the electrode width increases, the electric field strength above the electrodes decreases, creating gaps in the travelling electric field. Wider electrodes result in a larger gap. If the gap is large enough, a barrier is created because the twDEP force is too low for bacteria to cross. It is easier for larger cells to cross these gaps, reaching higher regions [69]. As shown in Figure 6E, a microelectrode track design with progressively larger gaps between the electrodes causes the twDEP force and negative DEP force that compensate for the gravitational force to decrease gradually along the microelectrode track. Thus, the microparticles could be trapped in given locations based on their physical properties.

Figure 7A demonstrates the implementation of twDEP for the trypanosomes separation from both human and mouse blood using a spiral array of electrodes. Thus, the difference in polarizability between cells and trypanosomes leads to their separation through opposing bi-directional movement. Small immotile red cells are repelled from the spiral array of electrodes by experiencing a negative dielectrophoretic force. In contrast, larger cells, such as pathogenic trypanosomes, are pushed toward the electrode array center by a positive force.

Since trypanosomes can produce considerable forces through their flagellum, they can propel themselves free of the electrode and experience positive attraction to the next one in the spiral. Finally, a pure sample of trypanosomes is concentrated in the middle of the spiral as non-motile cells remain fixed at the electrode they are first attracted to.

It is possible to separate particles inside a droplet containing a mixture of particles with twDEP method. The optimized twDEP frequency signal and the medium conductivity based on the dielectrophoretic properties of the particles that are to be separated and concentrated result in successful in-droplet separation and concentration [15]. After separating the particles towards both sides of the droplet, we can divide the mother droplet into two daughter droplets using the electrowetting-on-dielectric (EWOD) principle [102,104]. Fathi et al. have studied the separation of viable and non-viable yeast cells in a droplet via an electric field. In their method, a droplet is placed between parallel electrodes and a parallel plate with deposited electrodes. Cell separation by twDEP is performed with boomerang-shaped electrodes at the bottom. After the cells are separated, the upper electrodes cut the droplets using square shapes. According to the numerical simulation conducted in this study, the effect of twDEP on nonviable yeast cells is minor. In contrast, viable yeast cells are collected around the curvature of the electrodes. As a result, this method offers a significant advantage since it enables yeast cells to accumulate at the right corner of the droplet and ensures that they stay at the right daughter droplet after the droplet is cut [105]. Zhao et al. achieved an efficiency of over 92% for separating GP spores from glass beads using this method [103]. A system depicted in Figure 7B combines three complementary methods of DEP, twDEP, and Electrorotation (ROT) on a single, integrated chip. A microelectrode array chip is microfabricated on the silicon substrate, facilitating the non-uniform electric fields synthesis required for controlling, measuring, and characterizing mammalian cells [106]. TwDEP methods’ parameters have been reported in Table 3.
Figure 7(**A**) Trypanosome Enrichment and Detection in Blood using Counterflow Dielectrophoresis and spiral electrodes (Reprinted with permission from [70]. Copyright 2012, Springer Nature); (**B**) the combination of DEP, twDEP, and Electrorotation microchip to characterize and manipulate human malignant cells (Reprinted with permission from [106]. Copyright 2004, Elsevier).
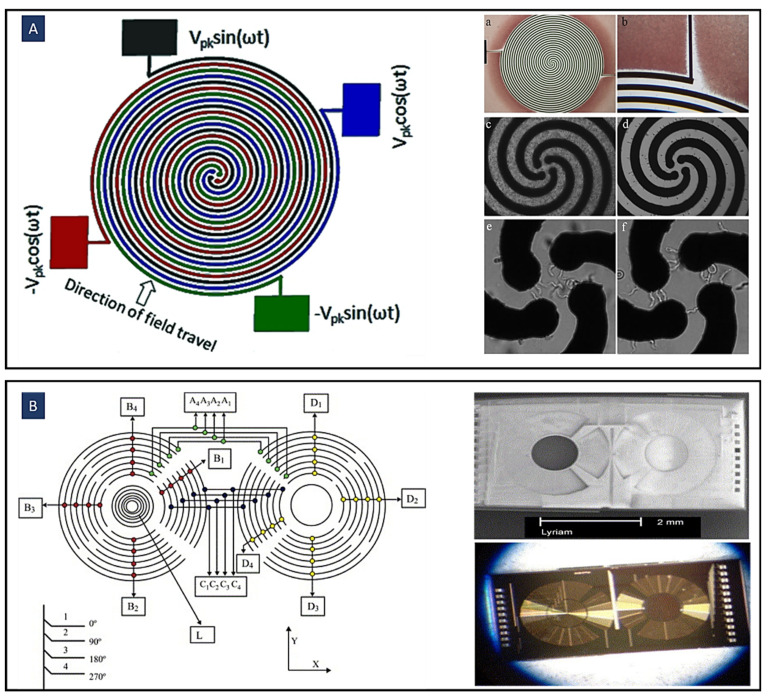



### 4.3. Time Varying Dielectrophoresis

In the traditional DEP, a continuous AC signal exerts a negative or positive force on the target particles. Time-varying DEP is another type of DEP that uses variable signal types (in frequency or amplitude) to rectify the limitations of the traditional DEP.

For example, in pulsed DEP (puDEP), the frequency of the signal changes between zero and a specific number in determined periods so that the DEP force changes between positive or negative and zero states. This method can be used to supply oscillating force, and by controlling the parameters of the time (frequency and task cycle), we can adjust its interaction with the fluid forces in the channel. In other words, this method is based on the competition between a fixed fluid drag and a discontinuous DEP force to influence separation.

In continuous DEP, we cannot separate medium-sized particles due tothe uniform dependency of the force on the size of the particle. PuDEP can help us sort out the particles with special sizes or dielectric properties in heterogeneous populations and the mixtures that include more than two kinds of particles. In Figure 8A, Cui et al. used this method to separate particles with intermediate size (5 μm) from the whole population (3, 5, and 10 μm). They used a pulsed sinusoidal signal to switch between negative and zero forces. Firstly, an nDEP force is turned on to block all the particles in the upstream area of the electrodes; afterwards, for a short time, the DEP force is turned off to release the unwanted particles downstream [107].

Furthermore, in some cases, continuous DEP can cause some problems and limitations while working with different particles and especially cells. For instance, positive DEP force can continuously attract cells toward the electrodes and affect cell motion. Thus, the cell behavior is notably different from that under short and temporary forces. Moreover, it can cause cell accumulation while working with samples having a high concentration of cells or particles. PuDEP is used to confront these problems and limitations by decreasing the force effect time and turning the force mode in special duty cycles. PuDEP can control cell deflection and avoid cell accumulation and adhesion to the electrodes. For example, in Figure 8B, Techaumnat et al. used this method to separate the red blood cells (RBCs) from PS beads (as rare cells) in a high concentration of RBCs according to the difference in their dielectric properties. Separation can be performed by choosing a signal frequency in which the target particles (cells) sense a strong pulsed DEP force and are deflected and displaced along the electrodes’ edge toward the channel wall in zig-zag trajectories, while other particles sense a negligible DEP force and move in the flow direction [12].

In Figure 8C, Hongjun Song et al. used puDEP to separate human mesenchymal stem cells (hMSC) and their differentiation progeny (osteoblasts). The optimal frequency for the separation generating the major difference is the pDEP regime. Their method is based on changes in the membrane structure and the morphology of the stem cells after differentiation. They achieved a collection efficiency up to 92% for hMSCs with a purity up to 84%, and the collection efficiency up to 67%, and the purity up to 87%, for osteoblasts [71].

For setting the time parameter in a pulsed signal to achieve a successful separation, characteristics of the device have to be considered. The following relationship gives the typical time scale for a particle that moves in the microchannel:(11)T=LUf
where *L* is the system characteristic length dimension (which usually is in the order of 10 μm) and $U_{f\;}$ is the flow velocity (which usually is in the range of 100 μm/s). Hence, the time scale (*T*) is usually between 0.1 and 1 s. Therefore, the DEP must be adjusted at this time scale to obtain an additional time parameter (in addition to the signal frequency) for controlling the separation process [107]. In addition, in this method, a minimum duty cycle of *D_T_* for the pulse signal is necessary for the particles to pass a transverse direction of *W_T_* inside the channel [12]. In this system, increasing the number of electrodes decreases the necessary duty cycle for the cells to pass a particular distance (*W_T_*). In addition, increasing the electrode tilt angle for the same voltage amplitudes and flow rates leads to lower cell velocity; hence, the duty cycle has to be increased [12]. For the same applied voltages and duty cycles, increasing the cell concentration leads to a lower cell velocity, and as a result, it decreases the separation efficiency. Moreover, it can cause cell chain formation, which alters the cell velocity, especially in higher voltages; consequently, a larger duty cycle must be utilized for good efficiency [12].
Figure 8(**A**) Using puDEP for separation of intermediated size particles (Reprinted with permission from [107]. Copyright 2009, Royal Society of Chemistry); (**B**) separation of red blood cells (RBCs) and PS particles according to the difference in their dielectric properties using slanted electrodes (Reprinted with permission from [12]. Copyright 2020, John Wiley and Sons); (**C**) separating human mesenchymal stem cells (hMSC) and their differentiation progeny (osteoblasts) based on changes happening in the stem cells’ morphology and the structure of the membrane during their differentiation (Reprinted with permission from [71]. Copyright 2015, Royal Society of Chemistry); (**D**) using frequency hopping to separate different sizes of PS particles. (**a**) Capture mode. (**b**) release mode (Reprinted with permission from [72]. Copyright 2019, Elsevier).
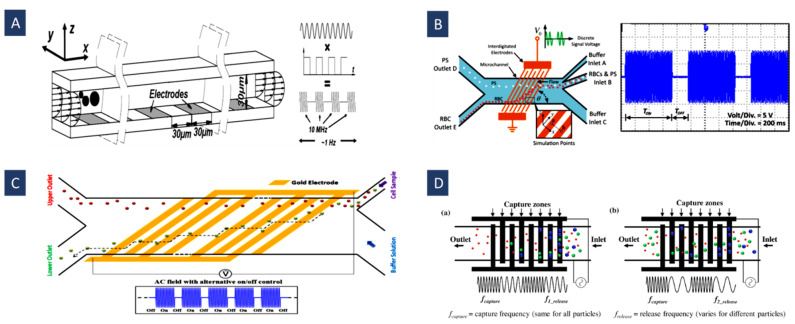



In another type of time-varying DEP, frequency hopping, two different frequencies are utilized in the specified periods to separate target particles. To this end, two different frequencies are utilized. The first frequency is chosen so that all the particles experience the maximum nDEP or pDEP force. Then, the second frequency is selected according to the particles’ size so that DEP force is weakened for the target particles causing them to be released from the DEP capture region. Hence, the application of the first frequency displaces all the particles to the field snare above the closest electrodes (nDEP) or in electrode edges (pDEP). The target particles are released from the DEP capture regions by tuning the second frequency. In this method, the second frequency decreases, approaching the critical frequency for the particles by reducing the flow rate. In Figure 8D, Modarres et al. used this method to isolate PS microspheres by two capture and release frequencies according to their sizes with a purity of 98.7%. They also investigated the feasibility of this method to separate CTCs (MCF7 cells) from RBCs with a separation efficiency of 82.2% [72].

Despite its benefits, puDEP has limited flexibility to optimize the trajectories for particles in a heterogeneous particle mixture. Particle trajectory can be obtained based on the particle size using a time-varying method in which the amplitude of the signal, rather than its frequency, is changed in a particular period (sawtooth signal). In this method, the competition between the DEP and the hydrodynamic forces leads to a forward motion for smaller particles. In comparison, larger particles have a back-and-forth motion without any forward motion. In other words, in one period, as voltage increases, particles move forward by DEP until they are trapped by the electrodes (in the least field intensity for nDEP). In the next voltage ramping, smaller particles sensing weaker DEP force travel further, while the larger ones remain trapped and have just back and forth movements. Modarres et al. introduced this method to optimize the trajectories of particles of different sizes [108].

### 4.4. Moving Dielectrophoresis (mDEP)

In moving DEP, excitable electrodes are activated sequentially, as shown in Figure 9. In this structure, successive switching of the electrodes on and off produces a moving electric field. The electric field moving speed is specified by the activation cycle of the electrodes. A moving DEP force is exerted by this moving electric field on the particles and displaces them in the desired direction. Hence, fluid flow is not necessary for particle separation.

Two parameters in the mDEP are controllable independently, the electric field frequency and the activation time between electrodes. In the traditional DEP, the frequency of the electric field is the single parameter of time that can be adjusted. Therefore, using it, we can only attain localized particle manipulation. In mDEP, the motion of the particles can be controlled in a moving electric field by a second-time parameter plus the frequency of the electric field.

This method is for easy separation of particles with various dielectric features or sizes. As mentioned before, in DEP, the electric field frequency can be set so that different particles experience different forces (positive or negative). When one electrode is connected to an AC signal, the particle undergoing pDEP is attracted to this electrode while the particle experiencing nDEP is repelled from it. When the particle experiencing nDEP gets to the adjacent electrode, the next electrode is connected to the AC signal. As a result, the second particle is repulsed further while the first one is still trapped in the first electrode [109].
Figure 9The moving DEP (mDEP) principle for particle manipulation; (**A**) the particles are attracted to the actuated electrode (pDEP) or repelled from it (nDEP); (**B**) particles that experienced pDEP stay in their place while particles that experienced nDEP move to the next activated electrode.
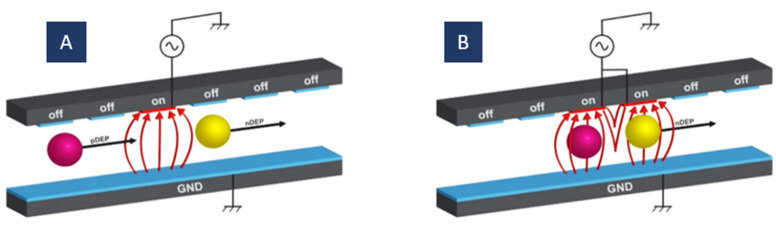



The distance between the particles can be determined by the change in the acting electrodes’ number. Furthermore, by applying the electric field frequency, the same as the desired particles’ relaxation frequency (CM-factor = 0), it is possible to separate a mixture of heterogeneous particles with various dielectric properties. While the electric field moves over the electrodes, the purpose particles experience zero dielectrophoretic force, but the others experience a pure dielectrophoretic force. Hence, the purpose particles stay stationary at the original position while the others are transported away. In this case, particle separation can also be performed based on particles’ speed. In other words, if the velocity of the particles is lower than the changing velocity of the electric field, particles will be left behind; otherwise, the particles will move forward [109].

This method is rather similar to twDEP but does not have some of its limitations. Like in twDEP, the fluid flow is not necessary for particle transporting; they are transmitted into the channel using just DEP force. In twDEP, the real part of the Clausius–Mossotti factor determines the particle levitation, and its imaginary part determines the translational movement of the particles. These two factors cannot be controlled independently. As a result, cells are separated and transported to the outlet just in the correct combination. In contrast, in mDEP, the cells are isolated and transported by just setting the real part of the CM factor and the time parameters independently [73].

As shown in Figure 10A, Kua et al. used this method to separate the viable and the nonviable Saccharomyces cerevisiae yeast cells. They set the frequency of the electric field so that the viable cells felt the pDEP and were trapped behind the activated electrodes, whilst nonviable cells experienced nDEP and were separated from the viable ones with a distance of three electrodes [73].

Pulsed DEP and moving DEP can be combined to provide the possibility of the contactless and addressable displacement of the particles (Figure 10B). To this end, two electrode layers can be used on the channel ceiling and bottom in opposite directions (vertical and horizontal). PuDEP is used by chopping the AC signal using a square wave. This chopped signal is applied to the vertical and the horizontal electrodes successively. Signal polarities are reversed in these two lines; as a result, induced dipoles on the particles are only continuous at the intersection of the electrodes. In other words, in a half pulse period, the upward/downward electrode is connected to the positive/negative potential. In the next half, this electrode is grounded, and the downward/upward electrode is connected to the negative/positive potential. This signal has the previous signal phase. Accordingly, similarly oriented dipoles are induced in the particles, and only at the intersection of the electrodes do particles experience continuous dipoles. In this case, moving DEP displaces the particles from one trap to another and makes the trapping addressable. Thibault et al. introduced this method for 3D manipulating particles in a suspension [74]. Time-varying methods’ parameters have been reported in Table 4.
Figure 10Using mDEP for cell/particle manipulation; (**A**) separation of viable and nonviable cells based on their different dielectric properties (Reprinted with permission from [73]. Copyright 2007, American Chemical Society(ACS) Publications); (**B**) using moving pulsed DEP for contactless and addressable particle manipulation (Reprinted with permission from [74]. Copyright 2013, Royal Society of Chemistry).
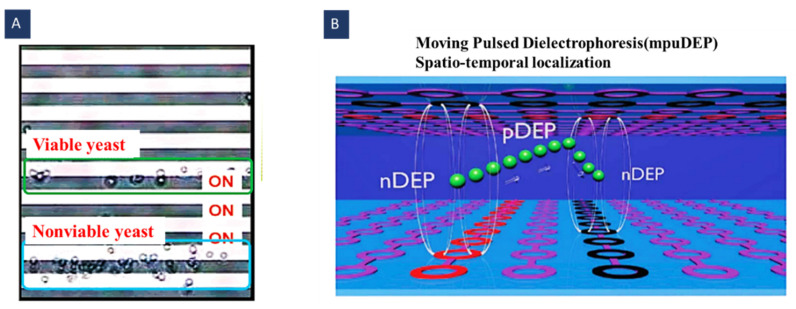



### 4.5. Field Flow Fractionation Dielectrophoresis (FFF-DEP)

Field flow fractionation (FFF) is an isolation method in which a field that is perpendicular to the flow direction is exerted on the flowing particles. This technique uses differences in the particles’ mobilities for separation. An applied field may be electrical, gravitational, magnetic, centrifugal, transverse flow through a semi-permeable membrane, thermal gradient, etc. [110]. FFF can separate a wide range of colloidal particles while maintaining high resolution; hence, it is unique from other separation methods. The principle of this method is generating a pressure-driven flow (parabolic flow) which carries the sample particles from the inlet toward the outlet of a channel. Particles with different characteristics (electrical, thermal, magnetic properties, etc.) are carried at different velocities following their positions in the stream. FFF is usually conducted in simple geometries (often rectangular microchannels); as a result, the behavior of the flows is predictable, and flow hydrodynamic can be calculated precisely [111,112,113].

FFF can be classified into different sub-methods according to the applied field nature, including sedimentation FFF, Centrifugal FFF, Gravitational FFF, flow FFF, Thermal FFF, magnetic FFF, electrical FFF, Symmetrical FFF, Asymmetric FFF, and dielectrophoretic FFF (DEP-FFF). Among these methods, DEP-FFF is a promising method using microfluidic technologies and microfabrication to separate a wide range of analytes. It is a non-invasive, label-free, simple in vitro technique for manipulating various Micro and Nanoparticles. It is usually utilized to fractionate the particles/cells according to their different dielectric properties and/or sizes using DEP force applied perpendicular to the flow direction [114].

FFF has three operation modes: normal [110], steric [115], and hyper-layer [116]. The normal FFF mode drives the elution of macromolecules and sub-micrometer particles. These macromolecules and particles are pushed toward the accumulation wall by the field. Consequently, their concentration increases with declining the distance from this wall (Figure 11A(a)). It leads to a concentration gradient that creates sample diffusion away from the wall [113]. The basis of this mode is the diffusion coefficient, so the final distribution of the separated particles is determined thermodynamically. In DEP-FFF, the normal mode contains nanoparticles, cell organelles, and molecules (like proteins) with a diameter below ~1 µm. The diffusion phenomenon can be neglected for bioparticles (like cells) owing to their large size (usually larger than ~1 µm) [114].

Steric mode FFF controls the elution of particles with sizes larger than ~1 µm [113]. In this mode, the influence of the Brownian diffusion on the microparticles is negligible. These particles usually gather in the accumulation wall and create a thin layer. Larger particles’ center penetrates the faster streamlines farther away from the accumulation wall. Hence, larger particles get out of the channel sooner than the smaller particles whose centers are closer to this wall (Figure 11A(b)). In this mode, the elution of the particles is governed by the following equation (Figure 11B) [114]:(12)$$\;\;\;\;\;F_{sedim}+F_{p-w}+F_{p-p}+F_{DEP}=0$$
where $F_{sedim}$ is the sedimentation force, $F_{p-w}$ is the contact force between the accumulation wall and the particle, $F_{p-p}$ is the interaction force among two adjacent particles, and $F_{DEP}$ is the dielectrophoretic force.

The hyper-layer mode is also used for particles larger than ~1 µm [113]. Particles in the steric mode are affected by the hydrodynamic lift forces; however, in the hyper-layer mode, its direction is opposite of the external force field direction. In hyper-layer mode, particles move away from the wall by the lift force. In addition to the size, other physical properties like deformability and shape, are influential on the separation (Figure 11A,C) [114,116]. In this mode, the particle center is pushed away to a distance larger than the particle radius. Furthermore, there is no contact between the particle and wall, and the particle feels a hydrodynamic lift force which is a function of its distance from the wall. For DEP-FFF particle elution in hyper-layer mode is described in Figure 11B [114].
(13)$$F_{sedim}+F_{lift}+F_{p-p}+F_{DEP}=0\;$$

In DEP-FFF, steric and hyper-layer can be utilized simultaneously for performing the separation in complicated mixtures of particles. In this case, particles with low crossover frequencies are trapped while hyper-layer DEP-FFF at higher crossover frequencies separates other particles with high discrimination. When DEP and sedimentation forces affect a particle group in the same direction, particles move toward the electrodes. In this place, steric hindrance and the low flow rate dominating near the chamber floor decelerate their movement. In this case, cell elution features are determined by the size of the cell and flexibility of the membrane, which influences hydrodynamic lift force in this situation [117,118]. On the other hand, if the DEP and sedimentation forces have opposite directions, particles are levitated into the fast streamlines and transported to the output more rapidly. Gascoyne et al. used this method to isolate tumor cells with efficiencies above 90% [112].

TwDEP can also be used in FFF for lateral displacement. Under the balance of dielectrophoretic levitation and gravitational forces, particles are positioned in the channel at different equilibrium heights; hence, they are transported at various velocities by the fluid. Simultaneously, a horizontal travelling wave dielectrophoretic force is subjected to the cells to detect them across the flow stream. In Figure 11C, Gasperis et al. used this method for cell separation [75]. FFF-DEP methods’ parameters have been reported in Table 5.
Figure 11(**A**) FFF operation modes. Different separation mechanisms for particles of various sizes. (**a**) Normal, (**b**) steric, and (**c**) hyper-layer mode (Reprinted with permission from [6]. Copyright 2010, Royal Society of Chemistry); (**B**) forces exerted on the particles in hyper-layer and steric modes in DEP-FFF. The microelectrode array applying The DEP force on the particles is positioned on the accumulation wall. DEP force in steric mode can be positive or negative. This figure depicts the pDEP operation in the steric mode (desired particles are attracted to the electrodes). In the other case, the wall with electrode array is positioned opposite the accumulation wall and negative DEP pushes the target particles towards the accumulation wall. In the hyper-layer case, the particles are levitated away from the electrodes by nDEP force (Reprinted with permission from [114]. Copyright 2021, Elsevier); (**C**) separation of different groups of cells using FFF-DEP along with twDEP (Reprinted with permission from [75]. Copyright 1999, Springer).
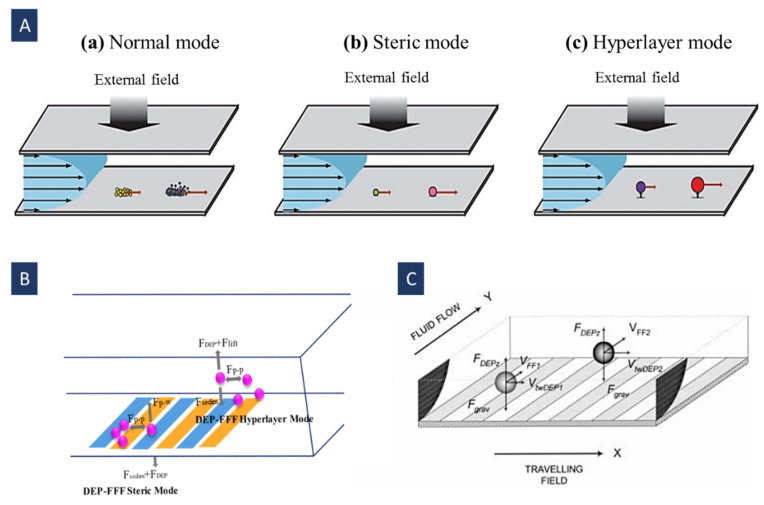

biosensors-12-00510-t002_Table 2Table 2Crossover frequency-based methods’ reported parameters.Refs(Method)Target ParticlesFrequency (Hz) Medium Conductivity (s/m)Particle Diameter (μm)Flow Rate (μL/min)Voltage (V_p–p_)[92](Single frequency)Live yeast cells-Dead yeast cells10 M
7.50.2256[64](Single frequency)MCF7—HCT116 3.2 M0.3
0.19[66](Multiple frequency DEP)Viable yeast cells-Nonviable yeast cells*F*_1_ = 5 K*F*_2_ = 5 M0.00287(nonviable)-8(viable)
5.7[65](Multiple frequency DEP)PS particles-Yeast cells50 K and 60 K (focusing)5 M (separation)0.1262-4-5 (PS particles)
10.4[119](Multiple frequency DEP)Viable yeast cells-Non-viable yeast cells60–90 K (focusing)5 M (separation)0.0607(nonviable)-8(viable)~03.39 and 4.38 (focusing)4.45 (separation)[97](UHF-DEP)Different types of Mesenchymal Stem Cells (MSCs)90 M0.0224



biosensors-12-00510-t003_Table 3Table 3TW-DEP methods’ reported parameters.Refs(Method)Target ParticlesFrequency(Hz)Medium Conductivity(s/m)Particle Diameter (μm)Flow RateVoltage (V_p–p_)[103](twDEP-flow) RBC-Liposome100–3 M0.1 m5.8 (RBC)1.5–4.6 (liposome)1.67 (±0.83)–42 (±10) μm/s2–16[76](Gradient twDEP)*S. cerevisiae* bacteria-*L. casei* bacteria 180 K10 m5–10 (*S. cerevisiae*)2–4 (*L. casei*)3 μm/s3.5[106](Gradient twDEP)Latex microparticles200 K10 m3, 6, 10, 20
8[120](Spiral twDEP)Trypanosomes-Mice RBCs-Human RBCs10–140 M

10 μL/min2–4[67](twDEP-EWOD)Ground pine spores-AS latex beadsGlass beads800 K100 K1 K120 m8 (Ground pine spore)5 (latex beads)8 (glass beads)10 μL/min15 (800 KHz)10 (100 K)120 (1 K)[121](twDEP-EWOD)Viable yeast cells-Non-viable yeast cells70 K150 m8 (viable)7 (non-viable)
4[101](Gradient twDEP)*S. cerevisiae* bacteria-yeast cells450 K
5–10 (*S. cerevisiae*)8 (yeast)
4[122](DEP-twDEP-ROT)Viable Daudi cells-Non-viable Daudi cellsViable NCI-H929 cells-Non-Viable NCI-H929 cells
78 m, 94 m12 (viable Daudi)18 (non-viable Daudi)14 (viable NCI-H929)20 (non-viable NCI-H929)850 μm/s
[69](twDEP)Leukocyte500 K30 m7
10[102](Gradient twDEP)*E. coli* bacteria-yeast cell100–350 K3 m, 10 m0.5 (*E. coli*)8 (yeast)
3.5–5
biosensors-12-00510-t004_Table 4Table 4Time varying methods’ reported parameters.Refs(Method)Target ParticlesFrequency (Hz) *F_S_*: Sine Signal Freq*F_P_*: Pulse Signal Freq*D_T_*: Duty CycleMedium Conductivity(s/m)Particle Diameter (μm)Flow Rate(μL/min)Voltage (V_p-p_)[107](puDEP)PS beads*F_S_* = 10 M*F_p_* = 2 (10-μm beads)*F_p_* = 1.05 (5-μm beads)*F_p_* = 0.3 (3-μm beads)
3, 5, 100.8312 (10 μm)20 (5 μm)20 (3 μm)[12](puDEP)RBCs—PS beads*F_S_* = 5 M*F_P_* = 1.25*D_T_* = 0.750.0257.9 ± 0.5 (RBC)10 (PS)2.412[71](puDEP)Stem cells—their differentiation progeny.*F_S_* = 3 M0.02
1.8–5.4 15.4[72](Frequency hopping) PS beadsRBC-MCF71*. F_capture_ = 1 M, F_release_ = 85 K, f_shift_ = 12*. F_capture_ = 1 M, F_release_ = 20 K, f_shift_ = 13*. F_capture_ = 1 M, F_release_ = 150 K, f_shift_ = 10.0283–5-10 (PS)9.14 (RBC)-24.34 (MCF7)0.66 (PS beads)0.83 (RBC-MCF7)20[73](Moving DEP)Viable-Nonviable Saccharomyces cerevisiaeyeast cells2 M0.03058 (viable cells)7 (nonviable cells)
9.3[74](Moving puDEP)PS beads *F_S_*: 50 K (pDEP)—2 M (nDEP) 1.5 < *F_p_*/*F_S_* < 50.0002

101*. Free-flowing 3 μm microspheres with 5 μm and 10 μm microspheres trapped. 2*. Free-flowing 3 μm and 5 μm microspheres with 10 μm microspheres are trapped. 3*. MCF7s were captured by a pDEP force, and RBCs were eluted by a weak nDEP force.
biosensors-12-00510-t005_Table 5Table 5FFF-DEP methods’ reported parameters.RefsTarget ParticlesFrequency(Hz) MediumConductivity(s/m)ParticleDiameter(μm)Flow Rate(μL/min)Voltage(V_p-p_)[75] MDA_435 cells—peripheral blood mononuclear (PBMN) cells45 K0.056
18[112] Breast tumor (MDA-MB)cells- peripheral blood15 K0.030
45002.8


## 5. Technical and Biological Challenges of DEP Approaches

Despite their benefits, DEP approaches studied here may have some challenges, such as technical and biological challenges. For instance, using metal electrodes can cause issues while working with these devices. The significant limitations of metal electrodes usually occur at frequencies under ~5 kHz [123]. Some of these limitations are the electrochemical generation of toxic species, electrolysis effects that lead to the generation of gas bubbles, as well as electro-osmotic induced fluid motion. In some techniques listed here, such as moving dielectrophoresis, electrolysis can be avoided because, unlike conventional DEP methods in which electrodes are switched on continuously during the operation, each electrode is switched on for just several seconds.

Bubble formation in liquid can affect electrical insulation and lead to electrode polarization and corrosion. Moreover, unless the suspension medium has a relatively low conductivity (e.g., <30 mS/m), electrode polarization effects can extend over 5 kHz [123]. Lower electrical conductivities can also mitigate the risk of Joule’s heating. The Joule’s heating effect usually causes harm to biological cells, change in the osmolarity and concentration of the DEP buffer owing to evaporation, and decreased sensitivity and selectivity. These common low conductivity culture suspensions will cause some restrictions in some cases, such as DEP-FFF technology. One possible solution is continuously cooling the channel during the operation of the device to prevent Joule’s heating effect [17,70,124,125,126].

Many of the mentioned issues can be avoided by shielding metal electrodes from the fluid in the main channel using other fabrication methods like liquid electrodes [119,127]. It can also be advantageous for laboratories without photolithography systems because liquid-electrode DEP devices can be fabricated by injection molding and hot embossing. It can also reduce the mass production cost [123,128].

Moreover, where planar electrodes affect the cells/particles inside a channel, DEP force may not be strong enough to affect targeted particles. It is more potent in areas near the edges of the electrodes and exponentially decays as it travels away from the electrodes. A solution is to utilize 3D electrodes; however, they still lag behind in increasing the DEP force magnitude. A stronger DEP force will lead to the separation at higher flow rates because the DEP force competes with the hydrodynamic force. So the particles move fast and the action time of the DEP force exerting on the particles decrease [78]. Furthermore, a strong DEP force may also help develop microchannels with larger cross-sectional areas. The larger cross-section areas will improve flow rates. In addition, they solve the clogging problems, which are usual in many devices after continuous use of DEP. One solution for enhancing the DEP in the microchannel is changing the dimension of the electrodes, which requires introducing new materials [70].

We may face an issue while working with cells because cells must remain viable in the medium during and after the operation. After the DEP analysis, careful preliminary experiments should avoid significant changes in cells’ biological features. Moreover, the medium should have a low electrical conductivity, as mentioned before. In most DEP experiments, the usual cell culture media (such as phosphate-buffered saline (PBS); Dulbecco’s Modified Eagle Medium (DMEM)) cannot be used due to their high conductivity. Instead, low-conductivity media or buffers need to be used as the suspending medium [129]. One of the most critical challenges in cell separation is that cells are dynamic particles with variable dielectric properties according to their size, morphology, membrane, cytoplasmic components, developmental/physiological stage, etc. Therefore, even similar cell types (like breast tumor cells at early or late cancer stages) may differ in DEP characteristics [43,74], and it can affect the separation efficiency in these procedures. Many scholars are currently working in this field to tackle this problem [130].

An important challenge in the fabrication of these devices is that high levels of knowledge and practical experience are usually needed. In addition, the fabrication procedures are usually expensive, especially in mass production. In fact, commercialization and mass production are complex and have some limitations. For example, materials like PDMS are convenient for laboratory-scale production, but scaling up to the industrial level is difficult and costly. Moreover, PDMS is known to dissolve gasses and hydrophobic compounds that can influence the environment of cell suspensions. Emerging fabrication techniques (e.g., 3D printing) may help to overcome these challenges in the future. Furthermore, the lifespan of these devices may be affected by the buildup of debris from cell suspensions, aggregates of reagents, and residue from biofluids. These aggregates can further decline throughput and performance, which can cause longer operating times and become challenging for rare cell isolation [131,132]. Another important point is the automation of these devices without the need for any human operators. Currently, most of these systems are stand-alone, human-operated, and need to be automated to reduce their operational cost and time, and make them easy to use.

## 6. Conclusions and Perspectives

In general, various methods have been utilized for particle/cell separation in recent years. DEP has been one of the most successful techniques due to its advantages, including high sensitivity and simple fabrication procedures. Here we provided a comprehensive discussion on the dielectrophoresis-based methods for cell and particle separation/manipulation and their different aspects. We can identify and separate different groups of cells or particles in a mixture by working on different signal parameters. These methods are only dependent on the biophysical features of the cells/particles, field frequency, and the suspension’s dielectric properties. In other words, separation can usually be done in simple microchannels.

Current usual approaches in cell isolation are affinity-based and depend on the biochemical markers exhibited by the cell surface. Many immune, stem, and rare cancer cells do not show identifying biochemical markers associated with basic cell functions like surface activation, differentiation lineage, and metastasis, respectively. Many scientists currently notice that the electrical methods are suitable for cell isolation and do not have many of the previous methods’ limitations. Compared with other separation techniques, electrical methods benefit is that cell modification by antibodies or adhesion to foreign substances is unnecessary. Hence, the cell activation by these probes or cell damage is avoided. The DEP force depends on the dielectric properties and size of the particle and the dielectric properties of the suspension. In addition, in an AC field, the DEP force changes as a function of the frequency.

As a result, there are many factors by which DEP force can be controlled and utilized for various applications, especially for separation purposes. Biophysical properties of the cell, such as electrical properties (membrane capacitance and cytoplasm conductivity) and geometric properties (cell size), are related to protein and gene expression. Consequently, these properties can be utilized in cell classification and cell status assessment.

Approaches listed here can be used in label-free and noninvasive cell isolation using their biophysical properties and inherent features. For example, the apoptosis state of the cells can be controlled by the change in the cell capacitance [133]. Consequently, different cells with different apoptosis states can be separated by this feature. Moreover, various cell types like pre-metastatic and metastatic cancer cells, different lineages of stem cells, and even cancer cells with different aggressiveness can be separated by these methods easily. We envision that current complex approaches in cell isolation be replaced by these easy-to-use electrical methods in the future.

## Figures and Tables

**Figure 1 biosensors-12-00510-f001:**
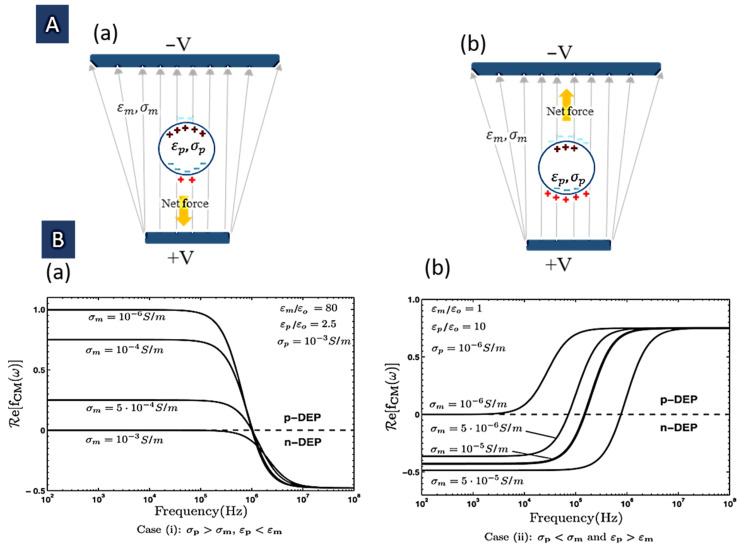
DEP force on an induced dipole in a non-uniform electric field. (**A**): (**a**) Positive DEP (p-DEP); (**b**) Negative DEP (n-DEP). (**B**): (**a**) DEP spectra of a dielectric particle with $\sigma_{p}>\sigma_{m\;},{\;\varepsilon }_{p}<\varepsilon_{m};\;$ (**b**) DEP spectra of a dielectric particle with $\sigma_{p}<\sigma_{m\;},$ $\varepsilon_{p}>\varepsilon_{m}$ (Reprinted with permission from [14]. Copyright 2011, John Wiley and Sons).

**Figure 3 biosensors-12-00510-f003:**
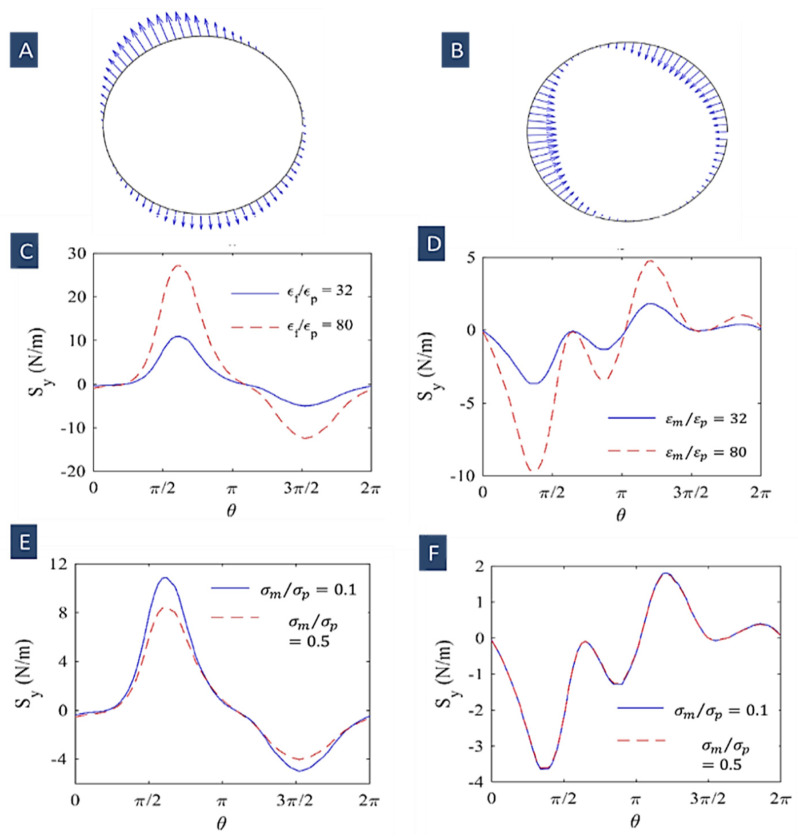
(**A**) low-frequency tensile stress on the cell surface in a non-uniform electric field for $\;\sigma_{p}=10^{-3}s/m$, $\varepsilon_{p}=2.5\varepsilon_{0}$, $\;\sigma_{m}=10^{-4}s/m\;$ and $\varepsilon_{m}=80\varepsilon_{0}$ (10 kHz); (**B**) high-frequency compressive stress on the cell surface in a non-uniform electric field for the same electrical properties (10 MHz) (Reprinted with permission from [84]. Copyright 2020, Springer Nature); (**C**) the magnitude of electric stresses (y-component) and effect of suspension permittivity in the frequency of 10 kHz; (**D**) the magnitude of electric stresses and effect of suspension permittivity in the frequency of 10 MHz; (**E**) the magnitude of electric stresses and effect of suspension conductivity in the frequency of 10 kHz; (**F**) the magnitude of electric stresses and the effect of suspension conductivity in the frequency of 10 MHz (Reprinted with permission from [84]. Copyright 2020, Springer Nature).

**Table 1 biosensors-12-00510-t001:** Electrode structures used in studied signal-based methods.

Refs	Electrode Types	Material	Methods
[64]	planar electrode (2 electrodes)	ITO	Crossover-based separation
[65]	Planar electrode	Ti/Pt	Crossover-based separation
[66]	Planar electrode (3 electrodes)	Au/Cr	Crossover-based separation
[67,68]	Top-bottom electrode array	Au/Cr	twDEP
[69]	Gradient electrodes		twDEP
[70]	Spiral electrodes	Au/Cr	twDEP
[12]	Slanted interdigitated electrodes	Al/Cr	Time varying DEP
[71]	Slanted interdigitated electrodes	Au	Time varying DEP
[72]	Simple interdigitated electrodes	Au	Time varying DEP
[73]	Top-bottom electrode array	ITO-Au/Cr	Moving DEP
[74]	Top-bottom electrode array	ITO	Moving-pulsed DEP
[75]	Simple interdigitated electrodes	Au	FFF-DEP

## Data Availability

Not applicable.

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
