# Peer review of "Signal-Based Methods in Dielectrophoresis for Cell and Particle Separation"

_biosensors, 2022, doi:10.3390/bios12070510_

Round 1
Reviewer 1 Report
This is a very well and clearly written review that covers the theoretical foundations of dielectrophoresis, the design and geometry of dielectrophoretic cells, frequency characterization and frequency methods, to biomedical applications. The paper is written in a readable manner with a minimum of formal typos and errors. Moreover, the review is written by leading experts in microfluidics and micromachines. For all these reasons, I recommend acceptance of the article for publication and in its present form.
Reviewer 2 Report
This is a very well written review covering DEP and its different operational modes. Authors have by and large done a great job of reviewing a dense literature. I recommend this for publication after following changes-
1) Please define scope of this paper. There are numerous review papers on dielectrophoresis. Describe how your review paper is different? How many papers have been reviewed? You could cite recent review articles and mention that DEP is quite interdisciplinary topic- from physics to applications. Your paper will focus on signal based DEP techniques that have been developed.
2) One key discussion missing is how the different signal based techniques compare with each other in performance ? (DEP force, throughput, efficiency)? Authors need to discuss that in main text. As a researcher reading this review article, which technique should I go for my DEP application?
3) Please consider making a table for different electrodes types/geometries to amend section 3.1.
4) Line 425- S in Subjection should be small letter not capitalized.
5) Line 974-975- References are missing.
6) Line 550, 555 and 558. Tab is missing for the paragraph beginning.
7) Line 991-995: References are missing for shielded electrode DEP. Please don't use another review paper to cite specific studies. It is not considered a good practice. Cite original work.
8) Section 5 is missing some key terms for technical challenges. Manufacturing challenges- chips manufacturing is not scalable. Each DEP device is super expensive. Second, throughput for applications like separation is extremely low. How many cells/hr can be sorted? Most of these systems are stand-alone human operated and lack end to end automation.
9) Section 5 biological challenges- Cells are inherently heterogenous. DEP can't really handle biological variability.
10) Line 1028- Not true. DEP devices are considered complex from manufacturing perspective. Please rephrase.
11) Line 1050- missing references.
12) Line 1054- missing references.
13) Line 558- missing references. Move citation ahead in the first line.
14) Line 856- beginning of the paragraph is missing a tab
15) Section 3 is not really tied well to the title of the paper. Can you rewrite it in a way that ties to different signal based approaches? For example, twDEP needs several series of electrodes-a specific electrode architecture.
16) Similarly, is there any different between different signal based techniques on cell viability?
